# Decoding the Versatile Landscape of Autophagic Protein VMP1 in Cancer: A Comprehensive Review across Tissue Types and Regulatory Mechanisms

**DOI:** 10.3390/ijms25073758

**Published:** 2024-03-28

**Authors:** Felipe J. Renna, Claudio D. Gonzalez, Maria I. Vaccaro

**Affiliations:** 1Instituto de Bioquimica y Medicina Molecular Prof Alberto Boveris (IBIMOL), CONICET, Universidad de Buenos Aires, Buenos Aires C1113AAC, Argentina; fjrenna@ffyb.uba.ar; 2Instituto de Investigaciones, IUC, Medicina Traslacional, Hospital Universitario CEMIC, Buenos Aires C1431FWN, Argentina; claudiodanielg@gmail.com

**Keywords:** cancer, autophagy, VMP1

## Abstract

Autophagy, a catabolic process orchestrating the degradation of proteins and organelles within lysosomes, is pivotal for maintaining cellular homeostasis. However, its dual role in cancer involves preventing malignant transformation while fostering progression and therapy resistance. Vacuole Membrane Protein 1 (VMP1) is an essential autophagic protein whose expression, per se, triggers autophagy, being present in the whole autophagic flux. In pancreatic cancer, VMP1—whose expression is linked to the Kirsten Rat Sarcoma Virus (KRAS) oncogene—significantly contributes to disease promotion, progression, and chemotherapy resistance. This investigation extends to breast cancer, colon cancer, hepatocellular carcinoma, and more, highlighting VMP1’s nuanced nature, contingent on specific tissue contexts. The examination of VMP1’s interactions with micro-ribonucleic acids (miRNAs), including miR-21, miR-210, and miR-124, enhances our understanding of its regulatory network in cancer. Additionally, this article discusses VMP1 gene fusions, especially with ribosomal protein S6 kinase B1 (RPS6KB1), shedding light on potential implications for tumor malignancy. By deciphering the molecular mechanisms linking VMP1 to cancer progression, this exploration paves the way for innovative therapeutic strategies to disrupt these pathways and potentially improve treatment outcomes.

## 1. Introduction

Neoplasia consists of a genetic disorder of cellular growth triggered by acquired or, less frequently, inherited mutations that affect a single cell and its clonal descendants. It is characterized by the uncontrolled proliferation of cells resulting in a growth pattern that lacks proper regulation. These cells lose responsiveness to physiological growth inhibitors and inducers, and the alterations become irreversible once the initiating stimulus subsides. A tumor is an abnormal mass or swelling of tissue resulting from the excessive and unregulated proliferation of cells. Tumors can be either benign or malignant. Cancer, and specifically the accompanying malignant tumors, poses a heightened risk to the organism due to its rapid growth and its potential to invade and destroy adjacent structures and spread to distant sites (metastasize). Numerous molecular changes drive cancer cells, enabling accelerated growth and metabolism [1].

Macroautophagy, or autophagy, is a physiological catabolic process that involves the degradation of various cellular components in the lysosome. This mechanism aids in maintaining cellular homeostasis by breaking down long-lived proteins and damaged organelles. Healthy cells leverage autophagy to regulate the organelle and protein turnover, reducing oxidative damage and preventing cancer initiation. Conversely, autophagy becomes advantageous in cancer cells with elevated metabolic demands and the need to survive under nutrient and oxygen deprivation. It allows cancer cells to adapt to their microenvironment and, additionally, it facilitates resistance to chemotherapy by mitigating cellular damage caused by such treatments [2].

Vacuole Membrane Protein 1 (VMP1), identified as a crucial autophagic protein induced during acute pancreatitis [3,4], has been implicated in pancreatic cancer progression [5]. It is also induced by chemotherapy agents, aiding cancer cells in resisting treatment [6,7]. However, VMP1’s role in various cancers is context-dependent, influenced by factors such as cellular context, tumor microenvironment, tumor stage, and the specific pathways involved. This review comprehensively examines the relevant literature, elucidating the intricate interplay between autophagy and cancer, as well as the diverse roles of VMP1 in different cancer types. The focus is on molecular mechanisms, connections to malignancy, interactions with micro-ribonucleic acids (miRNAs) in a cancer context, and instances of VMP1 gene fusion with other genes in cancer.

## 2. Role of Autophagy in Cancer

Autophagy exhibits a dual role in cancer development, with evolving research shedding light on the intricate mechanisms governing its impact on cancer initiation and development. While autophagy is acknowledged as a suppressor of tumor initiation, emerging evidence suggests its essential role in supporting uncontrolled cell growth and heightened metabolic activities within established tumors, resulting in a dependence on autophagy for tumor maintenance. The influence of autophagy extends beyond intrinsic functions within tumor cells to extrinsic effects in the surrounding stroma, influencing both tumor growth and drug resistance. The nuanced effects of autophagy are contingent on factors such as tumor stage, specific oncogenic mutations, and the cellular context, underscoring the complexity of its involvement in cancer progression [2].

The BECN1 gene, which encodes the autophagic protein Beclin-1, was the first autophagic gene to be described to play a role as a tumor suppressor. BECN1 loss was frequently found in cell lines from breast cancer and material from mammary tumors [8,9]. Moreover, partial deletion throughout the entire body resulted in tumor development, specifically in the lungs, liver, and lymphatic tissue, sparing other organs and tissues [10]. Another example of autophagic tumor suppression is Autophagy Related 7 (ATG7). The sole deletion of ATG7, without concurrent genetic alterations, induced tumor formation exclusively in the liver [11]. Notably, selective autophagy forms, such as mitophagy and pexophagy, actively contribute to tumor suppression by mitigating reactive oxygen species-induced stress [12,13,14]. Autophagy cargo receptors, exemplified by Sequestosome 1 (p62), play multifaceted roles in cancer, and their maintenance through autophagy-mediated degradation is identified as a key tumor-suppressive mechanism [2,11]. Importantly, the tumor-suppressive transcription factor Cellular Tumor Antigen p53 (p53) modulates autophagy in response to cellular stress, demonstrating a reciprocal relationship [15,16].

Regarding tumor progression, autophagy plays a complex role, as evidenced by high levels of autophagosomes in certain tumor tissues [17]. While the static nature of these readouts poses challenges in distinguishing autophagy induction from impaired turnover, preclinical studies highlight autophagy’s support of advanced tumor growth and metabolism following the activation of oncogenes or the inactivation of tumor suppressors [2,12,18]. In the context of oncogenic RAS-driven cancers, autophagy becomes essential to mitigating the increased demand for cellular energy and anabolic precursors, allowing sustained tumor development. Studies have revealed autophagy dependency in the progression of RAS-driven cancers, with its absence potentially enhancing early tumorigenesis, yet acting as a blockade in further cancer development [19]. This interplay extends to tumor suppressor genes—such as TP53, Phosphatidylinositol 3,4,5-Trisphosphate 3-Phosphatase, and Dual-Specificity Protein Phosphatase (PTEN)—whose deletion can alleviate the autophagy-dependent block of tumorigenesis, albeit not consistently, leading to fully established cancers [20,21]. Moreover, the type of oncogenic lesion can dictate the role of autophagy, as exemplified by lung tumors driven by Serine/Threonine-Protein Kinase Stk11 Homolog (LKB1) loss, where autophagy impairment becomes lethal upon concurrent ATG7 deletion [22].

The role of autophagy in cancer metastasis is multifaceted and remains contentious. Initial evidence suggested that autophagy promotes crucial pathways for efficient metastasis, including migration, invasion, resistance to cell death, adaptation to nutrient deprivation, and survival in foreign tissue microenvironments [2]. However, recent studies present a contrasting perspective, indicating that autophagy may restrict key steps in the metastatic cascade, particularly in controlling the emergence from dormancy and suppressing metastatic colonization and outgrowth [23,24]. Mechanistically, impaired turnover of autophagy cargo receptors, such as p62 and Next to BRCA1 gene 1 protein (NBR1), is identified as a key mediator of the metastatic phenotype in autophagy-deficient backgrounds [25,26]. Overall, the intricate and context-dependent roles of autophagy in cancer metastasis necessitate further investigation for therapeutic targeting.

Autophagy’s impact on the tumor microenvironment extends beyond tumor cells, influencing both the host stromal cells and the systemic milieu. Studies reveal that systemic autophagy inhibition, achieved through systemic ATG7 deletion in mice, can significantly enhance the regression of tumors driven by oncogenic Kirsten Rat Sarcoma Virus (KRAS) [27,28]. Autophagy in host cells, including pancreatic stellate cells, contributes to the anabolic rate of tumors, providing essential amino acids crucial for tumor growth and survival [29]. Notably, the inhibition of autophagy in one organ may affect the growth of a distant tumor, exemplified by liver-specific autophagy deletion releasing arginase I and limiting arginine availability for distant lung tumors [30]. Furthermore, stromal autophagy in cancer-associated fibroblasts (CAFs) plays a pivotal role in controlling protein secretion, impacting tumor progression through the modulation of growth factors and inflammatory cytokines [31]. Autophagy-dependent secretion, including the release of extracellular vesicles, influences intercellular communication within the tumor microenvironment and may be a potential target for cancer therapy [32]. Autophagy’s involvement in tumor immunity is also highlighted, affecting antigen presentation, immune cell trafficking, and responses to immune checkpoint blockade therapy [33]. These multifaceted interactions underscore the intricate role of autophagy in shaping the tumor microenvironment and present opportunities for therapeutic interventions targeting both tumor and stromal components.

## 3. Role of VMP1 in Autophagy

The characterization of the VMP1 gene has been conducted in the context of studying changes in gene expression in pancreatic acinar cells during acute pancreatitis. The VMP1 gene was discovered through the development of a copy DNA (cDNA) library of overexpressed genes in an experimental rat model of pancreatitis. The characterization of VMP1 revealed it to be a 406-amino-acid protein with six putative transmembrane domains, an isoelectric point of 6.28, and a molecular weight of 45.9 kDa. It was demonstrated that VMP1 is a protein whose expression is highly and rapidly induced in response to acute pancreatitis [3].

VMP1 is an autophagy-related protein whose expression, per se, induces autophagy in mammalian cells [4]. VMP1 interacts with Beclin-1, which is part of the Class III Phosphoinositide 3-Kinase Complex I (PI3KC3-C1) that enriches Phosphatidylinositol 3-Phosphate (PI3P) in the endoplasmic reticulum region from which the autophagosome emerges [34]. The interaction between VMP1 and Beclin-1 takes place at contact sites between the endoplasmic reticulum and the plasma membrane, which constitute initiation sites for autophagosomal biogenesis [35]. This interaction recruits the PI3KC3-C1 to the endoplasmic reticulum membrane, thereby allowing its action and the subsequent formation of the autophagosome [4,34].

VMP1 seems to play a crucial role in the closure and budding of autophagosomes. In cells lacking VMP1 (knock-out cells), autophagosomes exhibit a persistent state of openness and remain attached to the endoplasmic reticulum [36]. Additionally, VMP1 is implicated in the autophagosome membrane during the selective autophagy of activated zymogen granules (zymophagy) in an acute pancreatitis model [37]. Furthermore, VMP1 is identified as a mediator of mitophagy in the context of experimental acute pancreatitis [38].

Recent studies have revealed that VMP1 remains involved in autophagic flux until the autolysosome stage and is subject to regulation through ubiquitination [39]. Notably, VMP1 has been percieved to possess enzymatic functions as a phospholipid scramblase. However, the implications of this enzymatic activity in the autophagic mechanism are currently under investigation [40,41].

Figure 1 summarizes the involvement of VMP1 in the different stages of the autophagic process.

## 4. Role of VMP1 in Cancer

### 4.1. Relation between VMP1 and Malignancy

The available data in the bibliography highlight the multifaceted role of VMP1 in various aspects of cancer, encompassing initiation, metastasis, drug resistance, and resistance to cell death. Despite this, the relationship between VMP1 expression and cancer malignancies appears to be non-linear, with conflicting findings in different studies. We explored a classification based on cancer tissue types to better understand these different findings. Importantly, the proposed effects of VMP1 expression on malignancy across various cancer types are summarized in Table 1.

#### 4.1.1. Pancreatic Cancer

In the context of pancreatic tissue, VMP1 first gained attention within the framework of acute pancreatitis. In a normal pancreas, VMP1 expression is nearly undetectable, becoming activated under stress conditions such as pancreatitis, diabetes, or cancer transformation. Notably, the G12D mutation of the KRAS oncogene, a pivotal driver in pancreatic cancer initiation, has been linked to the induction of autophagy through VMP1 expression. This induction is mediated by the PI3K-AKT Serine/Threonine Kinase 1 (AKT1) pathway and involves a complex interplay between the transcription factor GLI Family Zinc Finger 3 (GLI3) and Histone Acetyltransferase p300 (EP300) [5].

Consistent with these molecular insights, VMP1 is found to be overexpressed in human pancreatic cancer compared to peritumoral tissues. This was evidenced by both direct observations from human biopsies [7] and bioinformatic analyses using the Gene Expression Profiling Interactive Analysis (GEPIA) platform that utilizes data obtained from human specimens [42]. Moreover, higher expression levels of VMP1 are associated with poorly differentiated pancreatic cancer in human specimens [7]. Experimental studies, such as those involving mice expressing G12D KRAS in the exocrine pancreas, have revealed that VMP1-mediated autophagy collaborates with the KRAS oncogene in pancreatic ductal adenocarcinoma (PDAC) initiation, significantly increasing pancreatic intraepithelial neoplasia (PANIN) formation in an autophagy-dependent manner [43].

Beyond initiation, the impact of VMP1 extends to therapeutic responses, as demonstrated in xenograft mouse models. The overexpression of VMP1 in these in vivo models results in increased resistance to the anti-tumoral agent gemcitabine [7]. Mechanistically, experiments performed in pancreatic cancer cell lines have shown that gemcitabine treatment induces VMP1 expression [6,44] through a pathway involving the Transcription Factor E2F1 (E2F1), and EP300 ultimately triggering autophagy [6].

The findings from these studies on pancreatic cancer underscore the promalignant role of VMP1 expression, implicating both tumoral promotion and chemotherapy resistance.

#### 4.1.2. Breast Cancer

Despite the comparatively lower number of studies investigating VMP1’s role in breast cancer compared to pancreatic cancer, existing data indicate an upregulation of VMP1 expression in breast cancer. A prospective study involving 95 patients with invasive breast cancer, employing next-generation sequencing, revealed a fivefold upregulation of VMP1 in triple-negative breast cancer [45]. Furthermore, an analysis of human data from The Cancer Genome Atlas (TCGA) and the Molecular Taxonomy of Breast Cancer International Consortium (METABRIC) demonstrated a significant correlation between elevated VMP1 levels and shorter survival periods in Receptor Protein-Tyrosine Kinase (HER2)-positive breast cancer. This suggests a potential role for VMP1 as a prognostic marker indicating a poor prognosis [46].

#### 4.1.3. Colon Cancer

Regarding colon cancer, the role of VMP1 in malignancy appears to be more nuanced compared to its roles in other types of cancer. On the one hand, research indicates that VMP1 promotes therapy resistance, similar to pancreatic cancer. Specifically, in colorectal cancer cell lines, the reduction in VMP1 has been shown to decrease viability in response to apoptosis inducers [47]. Furthermore, VMP1 expression is crucial for cell survival during photodynamic therapy in colon cancer cells. Mechanistically, Hypoxia-Inducible Factor 1-Alpha HIF-1α induces VMP1 expression, activating autophagy and promoting resistance to photodynamic therapy [48]. Additionally, elevated VMP1 expression in colon cancer cells can stimulate exosome secretion, facilitating resistance to 5-Fluorouracil (5-FU). This upregulation further amplifies the levels of the resistance protein Multidrug resistance-associated protein 1 (ABCC1) and the anti-apoptotic protein Apoptosis Regulator Bcl-2 (Bcl-2) [49].

On the other hand, VMP1 expression is found to be diminished in colon cancer, and its reduced expression correlates negatively with various aspects of tumor malignancy, such as proliferation and migration capacity. A study [50] revealed lower VMP1 expression in colorectal cancer human tissues compared to adjacent non-cancer tissues, and this reduced expression is associated with poorer patient survival. Mechanistically, VMP1 expression diminishes invasion and proliferation in colon cancer cells, which was confirmed in xenograft mouse models. The proposed molecular mechanisms were studied in colon cancer cell lines and involve the activation of the PI3K/Akt/Tight Junction Protein ZO-1 (ZO-1)/E-cadherin pathway. However, the same study reports that the downregulation of VMP1 significantly decreases resistance to 5-FU, aligning with the pro-therapeutic resistance role observed in the studies [47], [48], and [49]. In addition, the downregulation of VMP1 is found to increase the migration and invasion of CAFs through a pathway involving HIF-1α and miR-210 [51].

Upon integrating the studies on VMP1 in relation to colon cancer, experimental data consistently suggest that increased VMP1 expression promotes resistance to therapy, encompassing both chemical agents and photodynamic therapy. Notably, VMP1 appears to play a dual role, as its promotion of therapy resistance contrasts with its inhibitory impact on cancer spread, leading to decreased proliferation and invasion. Intriguingly, the regulation of VMP1 expression in cancer-associated fibroblasts differs from that in colon cancer cells. While HIF-1α induces VMP1 expression in colon cancer cells, it paradoxically decreases VMP1 expression in CAFs. This discrepancy may signify distinct roles of autophagy in response to hypoxia in these two cell types. Furthermore, the opposing effects of VMP1 expression on invasion capabilities in cancer cells compared to CAFs add another layer of complexity to its role in the tumor microenvironment.

#### 4.1.4. Hepatic Cancer

The role of VMP1 in hepatic cancer, particularly in hepatocellular carcinoma (HCC), is elucidated through two key studies that are consistent in that they both show that VMP1 expression is inversely associated with the tumor malignance [52,53]. In the study [52], VMP1 is identified as a direct and functional downstream target of miR-210. The expression of VMP1 is negatively correlated with the expression of miR-210 in HCC human tissues. Importantly, VMP1 levels are reduced under hypoxic conditions, and the downregulation of VMP1 by miR-210 is implicated in the mediation of hypoxia-induced HCC cell migration and invasion [52]. On the other hand, in [53], the authors evaluated the expression of VMP1 in human HCC specimens. Their findings revealed a significant downregulation of VMP1 in human HCC tissues, and this downregulation is closely correlated with multiple tumor nodes, the absence of capsular formation, vein invasion, and poor prognosis of HCC. Consistent with these clinical observations, the expression levels of VMP1 in HCC cell lines negatively correlate with metastatic potential. Importantly, in vivo studies using a mouse model demonstrated that upregulation of VMP1 was associated with the suppression of growth and pulmonary metastases of HCC [53].

The findings from these studies on hepatocellular carcinoma indicate a protective role of VMP1 expression in decreasing invasion and metastasis, and improving the prognosis.

#### 4.1.5. Gastric and Esophageal Cancer

VMP1 seems to play a pro-malignancy role in gastric cancer. Through analyzing the TCGA human dataset, VMP1 has been found to be a gene in which overexpression is significantly associated with the presence of lymph node metastasis [54]. Consistently, by analyzing human datasets, VMP1 has been identified as a hub gene that is differentially expressed in gastric cancer [55].

In the context of human samples of esophageal cancer analyzed through RNA sequencing, polymerase chain reaction (PCR), and Sanger sequencing techniques, a notable finding involves the fusion of VMP1 with Ribosomal Protein S6 Kinase B1 (RPS6KB1). This fusion event is observed in approximately 10% of esophageal adenocarcinoma cases and is notably associated with a marked decline in overall survival. Mechanistically, the RPS6KB1–VMP1 transcript encodes a fusion protein characterized by abnormal subcellular localization in cancer cell lines. Importantly, this fusion protein exhibits impaired functionality in inducing autophagy in cell lines and, concurrently, it significantly accelerates the growth rate of non-dysplastic Barrett’s esophagus cells (CP-A cell line) [56].

While VMP1 expression appears to be directly associated with a pro-malignant role in both gastric and esophageal cancer, it is crucial to highlight that, in esophageal cancer, the elevated expression involves an abnormal protein that comprises only the C-terminal of VMP1. This truncated protein has been demonstrated to be non-functional, thereby impairing autophagy. Consequently, the role of VMP-induced autophagy may differ between these two types of cancer. Importantly, a more detailed analysis of the RPS6KB1–VMP1 fusion gene will be performed in Section 4.3 of this review article.

#### 4.1.6. Ovarian Cancer

Two studies contribute valuable insights into the role of VMP1 in ovarian cancer; however, they present conflicting perspectives. In one study, the findings reveal an elevated expression of VMP1 in human ovarian tumor tissues when compared to normal ovarian tissues. Additionally, the downregulation of VMP1 in this context correlates with reductions in cell proliferation and invasion in ovarian cancer cell lines [57]. Conversely, in the other study, the authors demonstrate that the inactivation of the von Hippel–Lindau (VHL) tumor suppressor gene results in a HIF-1α/miR 210-dependent decrease in VMP1 expression. This downregulation of VMP1 is associated with enhanced cell migration in ovarian cancer cell lines [58].

These seemingly contradictory findings highlight the complex and multifaceted role of VMP1 in ovarian cancer progression. Notably, it is important to emphasize that different cell lines were used in the two studies. In [57], the authors measured VMP1 levels in six ovarian cancer cell lines, but only downregulated VMP1 and analyzed proliferation and invasion in those in which VMP1 was overexpressed (A2780 and OVCAR3). On the other hand, in [58], the authors worked with 3AO and SKOV3, which are the two cell lines with the least VMP1 expression, according to the data shown in [57]. Therefore, the role of VMP1 in ovarian cancer malignancy is likely related to the genetic context of each particular tumor, warranting further investigation to elucidate the underlying mechanisms and potential therapeutic implications.

#### 4.1.7. Glioma

There is a study that explores the role of VMP1 in the development and malignancy of glioma and glioblastoma [59]. In this study, the authors used public databases to compare VMP1 expression in glioma and glioblastoma with normal brain tissue. They found that VMP1 is overexpressed in glioma and glioblastoma. Moreover, they observed a gradual increase in VMP1 expression from glioma grade 2 to grade 4, confirming these results by applying immunohistochemistry to human glioma specimens. Through the analysis of public datasets and their own patient cohort (101 patients), the authors found that VMP1 expression is inversely associated with patient survival, suggesting VMP1 as a poor prognostic predictor in glioma. Mechanistically, using VMP1 knockout (KO) glioma cell lines, the authors found that VMP1 expression is necessary for cell proliferation and to prevent apoptosis and G2/M phase cell cycle arrest. They demonstrated that in VMP1 KO cells, the autophagic flux was significantly blocked. Additionally, VMP1 KO cells exhibited increased sensitivity to radiotherapy and chemotherapy agents. These findings were consistent with observations from the CGGA database, where patients with VMP1 overexpression showed lower survival rates after both chemotherapy and radiotherapy [59].

Although only one study has been published analyzing the role of VMP1 expression in glioma malignancy, it is important to note that this study involves the analysis of various online datasets, a cohort of patients from Servicebio (Wuhan, China), and VMP1 knockout glioma cell lines generated using CRISPR/Cas9 technology. The results obtained from these three different experimental models are consistent, demonstrating a pro-malignant role of VMP1 expression in glioma. This role involves autophagy induction, cell proliferation, and therapy resistance, and ultimately leads to a worse prognosis for patients.

**Table 1 ijms-25-03758-t001:** Proposed Impact of VMP1 Expression on Malignancy Across Various Cancer Types.

Type of Cancer	VMP1 Expression [Reference]	Effect of VMP1 in Malignancy	Pathway Involved	Reference
Pancreatic		Induction of Autophagy	G12D KRAS/PI3K/AKT1/GLI3/p300	[5]
	Overexpressed [7,42]	Induction by chemotherapy agents	E2F1/p300	[6]
		Resistance to chemotherapy agents		[7]
		Promotion		[43]
Breast	Overexpressed [45]	Bad prognosis		[46]
Colon	Downregulated [50]	Resistance to chemotherapy agents		[47]
			Exosomes/ABCC1/Bcl-2	[49]
		Resistance to photodynamic therapy	HIF-1α	[48]
		Better prognosis		[50]
		Less invasion	PI3K/Akt/ZO-1/E-cadherin	
		Less proliferation		
		Less migration and invasion in CAFs	HIF-1α	[51]
Hepatic	Downregulated [53]	Less metastasis	miR-210	[52]
		Less growth		[53]
		Less invasion		
		Less metastasis		
		Better prognosis		
Gastric	Overexpressed [55]	Favors metastasis		[54]
Esophagic		Promotion	RPS6KB1/VMP1 fusion	[56]
Ovarian	Overexpressed [57]	Cell proliferation		[57]
		Invasion		
		Less cell migration	VHL/HIF-1α/miR210	[58]
Glioma	Overexpressed [59]	Bad Prognosis		[59]
		Cell Proliferation		
		Less apoptosis		
		Chemotherapy resistance		
		Radiotherapy resistance		

In red: effects of VMP1 expression that promote malignancy. In green: effects of VMP1 expression that attenuate malignancy.

### 4.2. Relation between VMP1 and MiRNAs in Cancer

MiRNAs constitute a category of small non-coding RNAs that regulate gene expression post-transcriptionally by interacting with messenger ribonucleic acid (mRNAs). Alterations to miRNAs play a role in the development and advancement of various human cancers. As a result, examining the expression patterns of miRNAs in human cancers is linked to aspects such as cancer identification, staging, progression, and response to treatments. Indeed, variations in both the quantity and quality of miRNAs are implicated in the initiation, progression, and metastasis of cancer [60].

Three miRNAs—miR-21, miR-210, and miR-124—were found to be associated with VMP1 expression. All of these miRNAs have been reported to play significant roles in the pathophysiology of cancer. In the following paragraphs, we will delve into the relationships between VMP1 and each of these miRNAs, exploring their connections with the mechanisms underlying cancer malignancy. Figure 2 schematically illustrates the relationships between VMP1 and these miRNAs in the context of cancer pathophysiology.

#### 4.2.1. MiR-21

MicroRNAs (miRNAs) initially emerge within lengthy transcripts referred to as primary microRNAs (pri-miRNAs). Several pri-miRNAs also possess protein-coding attributes, thus undergoing processing both as pri-miRNAs and as pre-mRNAs, encompassing 5' capping, splicing, and polyadenylation [61].

Among the initially characterized pri-miRNAs stands miR-21 [62]. MiR-21 belongs to an uncommon category of miRNAs, as it is situated near the 3'-untranslated region (3'-UTR) of a coding gene. Precisely, miR-21 is positioned immediately downstream of the VMP1 gene. Since the polyadenylation of the VMP1 transcript occurrs before the miR-21 hairpin, it was not initially deemed to contribute to its expression. Instead, the primary miR-21 transcript was found to commence within intron 11 of VMP1, resulting in a 3.4 kb capped, polyadenylated, and unspliced transcript [62]. Other miR-21 promoters and primary transcripts have also been identified within the terminal intronic regions of VMP1 [63,64]. Notably, an alternatively polyadenylated isoform of VMP1 was identified—termed VMP1–miR-21—initiating from the coding VMP1 gene and extending into the 3'-UTR, encompassing the miR-21 hairpin. VMP1–miR-21 is widely expressed in various cell lines and is processed by Drosha, indicating that alternative polyadenylation of VMP1 may give rise to a read-through VMP1–miR-21 fusion transcript [65].

MiR-21 is prevalently heightened in cancer [60]. Multiple mechanisms contribute to elevated miR-21 levels, including genomic locus amplification at 17q23 in various solid tumors [66]. Moreover, miR-21 expression is induced by diverse cancer-associated pathways such as hypoxia, inflammation, Activator Protein 1 (AP-1), and steroid hormones [63,67,68,69,70]. Elevated miR-21 expression correlates with increased growth and decreased apoptosis in numerous cell culture and animal models. Notably, heightened miR-21 alone has been demonstrated to induce a pre-B malignant lymphoid-like phenotype in animal models, emphasizing its role as a genuine oncogene [71]. Conversely, the deletion of miR-21 in mouse cancer models diminished tumor formation, underscoring its significance in cancer biology [72,73].

A negative correlation between the expression of miR-21 and VMP1 in colorectal cancer was demonstrated [74]. The activation of VMP1 transcription, either through small activating RNA (saRNA) or the transcriptional activator GLI3, leads to a reduction in miR-21 expression. Additionally, this study identified a regulatory loop involving miR-21 and VMP1. Although miR-21 was not described to target VMP1 mRNA, it inhibits the activation and nuclear translocation of Transcription Factor EB (TFEB), which can activate VMP1 transcription. Moreover, the authors showed that, when VMP1 expression is stimulated, miR-21 expression decreases, reducing migration and invasion, and enhancing sensitivity to 5-FU in colorectal cancer cells [74].

#### 4.2.2. MiR-210

The reviewed research works collectively indicate a significant association between miR-210 and VMP1 in the context of cancer, particularly lung adenocarcinoma, ovarian cancer, HCC, and colorectal cancer. All the reviewed works consistently show that miR-210 targets VMP1, decreasing its expression [51,52,58,75,76].

In lung adenocarcinoma cells, Tissue Inhibitor of Metalloproteinases-1 (TIMP-1) induces miR-210 upregulation through a CD63 Antigen (CD63)/PI3K/AKT/HIF-1-dependent pathway. This induction is associated with increased angiogenesis and poor prognosis in lung cancer patients, suggesting a pro-tumorigenic role for the TIMP-1/miR-210 axis. Among the other downstream targets of miR-210, VMP1 is downregulated in the presence of TIMP-1 [75]. In ovarian cancer, VHL inactivation promotes cell migration by stabilizing HIF-1α, upregulating miR-210, and diminishing VMP1 expression [58]. In HCC, hypoxia-induced downregulation of VMP1 by miR-210 mediates HCC cell metastasis. Moreover, in this context, a negative correlation between VMP1 expression and miR-210 levels is suggested [52]. It was reported that miR-210 is frequently upregulated in colorectal cancer tissues and correlates with large tumor size, lymph node metastasis, advanced clinical stage, and poor prognosis. MiR-210 promotes the migration and invasion of colorectal cancer cells and is induced by hypoxia, mediating hypoxia-induced metastasis. VMP1 was identified as a direct and functional target of miR-210 in colorectal cancer [76]. Consistently, it was demonstrated that HIF-1α regulates the migration and invasion of CAFs by upregulating miR-210, which downregulates VMP1 [51]. Importantly, the authors showed that miR-210 specifically targets the 3'UTR of VMP1. Moreover, the HIF-1α/miR-210/VMP1 pathway is implicated in the migration and invasion of CAFs in colorectal cancer, with potential implications for colorectal cancer metastasis [51].

Collectively, these findings highlight the intricate involvement of miR-210 and VMP1 in various cancer types, emphasizing their roles in cancer progression, metastasis, and poor prognosis. The regulatory pathways developed involve TIMP-1, VHL, and HIF-1α. In all cases, VMP1 downregulation was associated with a pro-malignant phenotype.

#### 4.2.3. MiR-124

MiR-124 exhibits varied roles in different cancer types, with upregulation observed in Acute Lymphoblastic Leukemia (ALL), contributing to glucocorticoid resistance by suppressing the Glucocorticoid Receptor (NR3C1) [77]. In colorectal cancer, miR-124 is downregulated, and its forced overexpression inhibits cell migration, invasion, and glycolysis through the Hepatocyte Nuclear Factor 1 Homeobox A (HNF1A)-antisense RNA 1 (AS1)/miR-124/Myosins of Class VI (MYO6) axis [78]. In gastric cancer, circular RNA HIPK3 (circHIPK3) downregulates miR-124, impacting patient survival [79], while in hepatocellular carcinoma, decreased hsa-miR-124-3p correlates with larger tumors and poorer survival [80]. Furthermore, miR-124 downregulation in breast cancer is associated with poor clinical outcomes [81] and, in lung cancer, methylation of miR-124 loci is linked to reduced survival [82].

Recently, it was reported that VMP1 is upregulated in older goats and found to be regulated by miR-124a. MiR-124a acts as a regulator of VMP1, influencing myoblast proliferation, autophagy, and apoptosis through the PI3K/Akt/Mammalian Target of Rapamycin (mTOR) pathway. That study provides valuable insights into the molecular mechanisms governing myoblast behavior, with potential implications for understanding muscle fiber growth in goats [83].

### 4.3. VMP1 Fused with Other Genes in Cancer

Frequently occurring somatic changes in cancer involve chromosomal aberrations that lead to the amalgamation of distinct genes. Initially identified in hematologic malignancies, these alterations have emerged as significant drivers of sarcomas and tumors originating from epithelial tissues. The common result of gene fusions is the combination of coding segments from two disparate genes, resulting in the creation of a chimeric protein endowed with novel properties. Nevertheless, instances of fusions exclusively involving the exchange of regulatory regions have also been documented [84].

In the existing body of literature, VMP1 has been known to fuse with various genes across diverse cancer types. The most frequently observed fusion partner of VMP1 is RPS6KB1, identified in multiple cancer types. Additionally, other genes, such as Clathrin heavy chain 1 (CLTC) and AC099850.1, have been found to exhibit fusion with VMP1. This section will comprehensively examine each of these fusion genes, exploring their potential implications in malignancy and delving into the specific types of cancers in which these fusions have been documented.

RPS6KB1 and VMP1 are neighboring genes located on the long arm of chromosome 17 at position 23. The fusion between VMP1 and RPS6KB1 was reported for the first time in 2011 [85]. In this work, the authors described a RPS6KB1/VMP1 fusion transcript that is the product of a tandem duplication and is present in breast cancer samples. Normally, the VMP1 gene is upstream to RPS6KB1. The tandem duplication changes this localization, setting the 5′ part of RPS6KB1 and the 3′ part of VMP1. The data reveal notable patterns in the predicted fusion junctions at the RPS6KB1 and VMP1 loci. Specifically, a majority of the predicted fusions at the RPS6KB1 locus involve exon 1, with a smaller number using exon 4, occurring within the larger introns, particularly intron 1 (the largest) and intron 4 (the second largest). Similarly, the most common predicted fusion point in the VMP1 gene involves exon 8 and is located within the largest intron (intron 7). These observations suggest that random somatic DNA breakage across the genes might be the mechanism of fusion formation [85].

In the breast cancer cell line MCF-7, a fusion between exon 2 of RPS6KB1 and exon 11 of VMP1 was found [85]. This observation was validated in the study [46], which explored RNA-Seq data from 45 cell lines. Importantly, the authors made the observation that this cell line in particular is HER2-negative. In [85], the authors found that, of 70 primary breast tumors in Singaporean patients, 22 were positive for the RPS6KB1/VMP1 fusion (31.4%). On the other hand, through the analysis of RNA-Seq data from TCGA and other studies [86,87,88], it was found that the fusion was present in only 5 of 1724 breast tumors (0.29%) [46]. The authors suggest that this discrepancy could be related to the ethnicity of patients (most of them of European descent) or to the method that was used. Importantly, of these five tumors positive for RPS6KB1/VMP1, only one was HER2-positive. Furthermore, the authors observed a notable enrichment of VMP1 fusion transcripts—not only in conjunction with RPS6KBY—particularly within HER2-positive tumors [46]. Despite the restricted size of the population analyzed in [85], the authors identified a potential direct association between the presence of RPS6KB1/VMP1 and an unfavorable prognosis for the patients.

In esophageal adenocarcinoma, the RPS6KB1/VMP1 fusion has been found in around 10% of cases [56]. In these cases, the fusion involved exon 1 of PRS6KB1 and exon 7 of VMP1. Notably, patients with esophageal adenocarcinoma who displayed the RPS6KB1/VMP1 fusion experienced markedly diminished overall survival compared to those without the fusion, which is consistent with [46]. Mechanistically, the RPS6KB1/VMP1 transcript encoded a hybrid protein that markedly accelerated the proliferation of non-dysplastic Barrett’s esophagus cells. In contrast to the conventional VMP1 protein, the RPS6KB1–VMP1 fusion displayed abnormal subcellular positioning and demonstrated diminished efficacy in initiating autophagy [56].

Other VMP1 gene fusions, where VMP1 is located in the 3' portion, have been identified in breast cancer, including CLTC/VMP1 [89] and AC099850.1/VMP1 [90]. Notably, CLTC/VMP1 is reported to be out of frame [89,91].

After exploring the mechanistic implications of these fusions in cancer malignancy, two hypotheses have emerged. Firstly, the RPS6KB1/VMP1 fusion is suggested to result in a dysfunctional protein incapable of inducing autophagy in esophageal adenocarcinoma [56]. Secondly, an intriguing explanation links to the expression levels of the oncomiR miR-21. It was found that the tandem duplication giving rise to RPS6KB1/VMP1 in breast cancer also leads to a duplication of miR-21 [85]. Correspondingly, in the study [92], it was demonstrated that there was significant overexpression of miR-21 in tumors positive for 3' VMP1 fusions [92]. Notably, the impact of gene fusion on tumor malignancy appears independent of VMP1 protein expression. This effect is observed in out-of-frame fusions, as well as truncating fusions and fusions involving non-coding regions, such as intronic sequences or UTRs.

## 5. Conclusions and Perspectives

Autophagy exhibits a dual role in cancer, acting as both a suppressor of tumor initiation and a promoter of advanced tumor growth and metabolism within established tumors. The influence of autophagy extends to the tumor microenvironment, affecting not only cancer cells, but stromal cells, and indeed the systemic milieu. The role of autophagy in cancer metastasis remains controversial, with studies presenting contrasting perspectives on its promotion and restriction of key steps in the metastatic cascade. Importantly, the role of autophagy in cancer malignancy can be affected by different factors such as tumor stage, oncogenic mutations, and cellular context [2].

VMP1 is recognized as an autophagy-related protein that induces autophagy in mammalian cells, interacting with Beclin-1 and playing a crucial role in autophagosomal biogenesis [4,34]. VMP1 is implicated in various forms of selective autophagy, including mitophagy and zymophagy, and is involved in autophagic flux until the autolysosome stage [37,38,39]. The role of VMP1 in cancer is explored across different tissue types, revealing a complex and context-dependent relationship. In pancreatic cancer, VMP1 is upregulated under stress conditions and associated with *KRAS* oncogene activation, playing a pro-malignant role in initiation and chemotherapy resistance [5,6,7,42,43]. Consistently, in glioma, VMP1 expression is directly associated with malignancy and poor prognosis, promoting cell proliferation as well as resistance to chemotherapy and radiotherapy [59]. In hepatocellular carcinoma, a protective role of VMP1 has been found, since VMP1 expression decreases tumor spread and improves the prognosis [52,53]. In breast, colorectal, gastric, esophagic, and ovarian cancer, conflicting findings exist regarding the relationship between VMP1 expression and malignancy, highlighting the need for further investigation into the genetic context of each tumor.

Three miRNAs—namely, miR-21, miR-210, and miR-124—have been identified as key players in regulating VMP1 expression across various cancer types. These miRNAs actively participate in the intricate molecular network that governs cancer progression, metastasis, and therapy resistance [72,73,80,93]. Notably, among these miRNAs, miR-210 exhibits the most direct association with VMP1 expression. In several cancer types, miR-210 has demonstrated its ability to downregulate VMP1, contributing to pro-malignant outcomes such as angiogenesis, metastasis, and an unfavorable prognosis [51,52,58,75,76]. Interestingly, miR-124a also targets VMP1, downregulating it. However, although it has only been tested in goat myoblasts until now, the downregulation of VMP1 by miR-124a is associated with less proliferation and more apoptosis [83], while its downregulation by miR-210 is associated with promalignant effects. It is another important example of how VMP1’s role in the malignancy of cancer cells can be totally affected by the genetic context. On the other hand, the connection between VMP1 and miR-21 is primarily attributed to genomic proximity. The VMP1–miR-21 fusion transcript is observed in various cell types, establishing a compelling association between these two molecules [65]. This intricate interplay between miRNAs and VMP1 underscores their significance in the complex landscape of cancer biology.

Additionally, different gene fusions involving VMP1, particularly the *RPS6KB1*/*VMP1* fusion found in breast cancer and esophageal adenocarcinoma, have been found. The *RPS6KB1*/*VMP1* fusion is associated with an unfavorable prognosis and dysfunctional autophagy induction, adding another layer of complexity to the understanding of VMP1 in cancer [46,56,85].

Leveraging the insights gleaned from the intricate regulatory network involving VMP1 in diverse cancer types, the modulation of VMP1 presents a promising avenue for therapeutic intervention. By understanding the molecular mechanisms through which VMP1 is intricately linked to cancer progression, novel therapeutic strategies could be devised to disrupt these pathways and potentially improve treatment outcomes. Precise modulation of VMP1 could emerge as a tailored approach for addressing specific cancer types, offering a targeted and nuanced therapeutic option in the broader landscape of cancer treatment.

Nevertheless, there are several challenges that must be addressed regarding the targeting of VMP1. Firstly, the lack of an experimentally confirmed structure of VMP1 poses a significant limitation for the development of new drugs specifically aimed at targeting this protein. Secondly, there are currently no drugs developed to specifically target VMP1. Thirdly, since VMP1 is an essential autophagic protein, blocking its function may lead to unacceptable adverse effects. Finally, it would be essential to develop methods for monitoring VMP1 or autophagic function in treated patients.

Despite these challenges, recent discoveries of two posttranslational modifications—ubiquitination [39] and palmitoylation [94]—have shed light on potential avenues for modulating VMP1 function. These modifications appear to regulate VMP1 function, suggesting that modulating these modifications could offer a viable method for tightly regulating VMP1 function in tissues where these modifications are more active, while minimizing adverse effects in other tissues.

Therefore, further in-depth research in these areas holds significant promise for the future of cancer treatment, offering potential solutions to overcome the challenges associated with therapeutic resistance, paving the way for more effective and tailored therapeutic strategies.

## Figures and Tables

**Figure 1 ijms-25-03758-f001:**
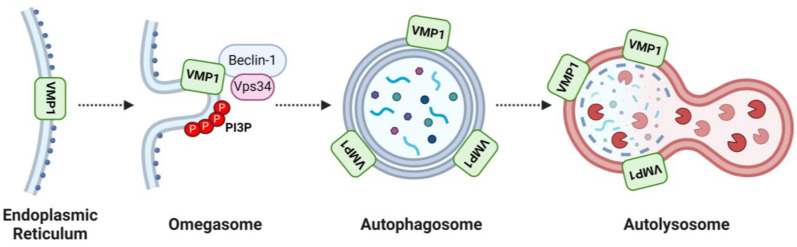
Involvement of Vacuole Membrane Protein 1 (VMP1) in Different Stages of the Autophagic Process. This figure schematically illustrates the role of VMP1 in various stages of autophagy. VMP1, a transmembrane protein synthesized in the endoplasmic reticulum, plays a crucial role in orchestrating autophagy. Initially, VMP1 interacts with Beclin-1, facilitating the recruitment of Class III Phosphoinositide 3-Kinase Complex I (PI3KC3-C1) and the enrichment of the omegasome with phosphatidylinositol 3-Phosphate (PI3P). Subsequently, VMP1 remains present throughout the autophagic process, spanning from the formation of the autophagosome to its fusion with lysosomes, forming the autolysosome. Vps34: Phosphatidylinositol 3-kinase VPS34. Figure created with BioRender.com.

**Figure 2 ijms-25-03758-f002:**
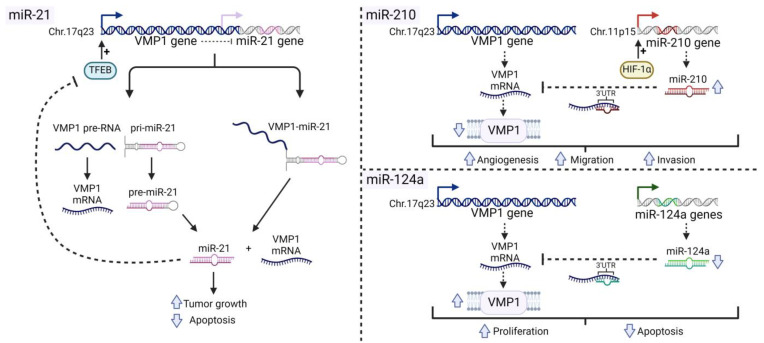
Relationships Between VMP1 and micro-Ribonucleic Acids (miRNAs) in Cancer. This figure schematically illustrates the relationships between VMP1 and miR-21, miR-210, and miR-124a, and their effects on cancer malignancy. MiR-21 is encoded downstream of the VMP1 gene. These genes can be transcribed separately, each being modulated by different factors, or they can produce a single transcript named VMP1-miR-21, giving rise to both VMP1 and miR-21. MiR-21 is known to promote tumor growth and inhibit apoptosis in various types of cancer. Moreover, VMP1 transcription can inhibit miR-21 expression, while miR-21 can inhibit Transcription Factor EB (TFEB)-mediated induction of VMP1 transcription. MiR-210 targets the 3′-untranslated region (3′UTR) of VMP1 messenger ribonucleic acid (mRNA), leading to the inhibition of VMP1 expression. Overexpression of miR-210 in cancer is associated with VMP1 downregulation and increased angiogenesis, migration, and invasion. MiR-124 also targets the 3′UTR of VMP1 mRNA, resulting in the inhibition of VMP1 expression. In this case, these effects lead to increased proliferation and decreased apoptosis. Figure created with BioRender.com.

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
