# Peer review of "Decoding the Versatile Landscape of Autophagic Protein VMP1 in Cancer: A Comprehensive Review across Tissue Types and Regulatory Mechanisms"

_ijms, 2024, doi:10.3390/ijms25073758_

Round 1

Reviewer 1 Report

Comments and Suggestions for Authors

The manuscript presents a thorough overview of the role of Vacuole Membrane Protein 1 (VMP1) in various cancers and provides valuable insights into its involvement in cancer progression and resistance to therapy. The authors effectively explain the dual nature of autophagy in cancer, emphasizing its ability to both suppress malignant transformation and promote tumor progression and resistance to treatment.

This review summarizes the existing literature to highlight the role of VMP1 in regulating autophagy and maintaining cellular homeostasis. In particular, the discussion of the association of VMP1 with pancreatic cancer, especially its correlation with the KRAS oncogene, sheds light on its significant impact on disease pathogenesis and therapeutic response. Furthermore, the extension of this study to other cancers such as breast cancer, colon cancer and hepatocellular carcinoma underscores the broad importance of VMP1 in different tissue contexts.

In addition, the review addresses the regulatory mechanisms underlying VMP1 function in cancer, including its interactions with micro-ribonucleic acids (miRNAs) and gene fusions with ribosomal protein S6 kinase B1 (RPS6KB1). These findings contribute to a deeper understanding of the complex molecular networks that drive cancer progression and offer potential avenues for the development of innovative therapeutic strategies.

However, there are areas where the manuscript could be improved:

·        Inclusion of information on the role of VMP1 in glioma would increase the scope of the review, considering its importance in the broader landscape of cancer research.

·        Restructuring Table 1 to separate "VMP1 expression" from "effect of VMP1 in malignancies" in two separate columns for each cancer type would improve clarity and intuitiveness for readers.

·        A more detailed discussion of the potential limitations or challenges associated with targeting VMP1 for therapeutic interventions would provide a more balanced perspective on the clinical implications.

Overall, this review provides a comprehensive overview of the role of VMP1 in cancer biology and provides potential starting points for future research focused on the development of targeted therapeutic approaches to improve cancer treatment outcomes.

Comments on the Quality of English Language Please correct a grammatical error in line 16.

Reviewer 2 Report

Comments and Suggestions for Authors

Thank you for submitting your review article for potential publication in IJMS. Here are my suggested comments as below:

  1. 1. The review article lacks sufficient grounding in references, particularly regarding in vivo and/or clinical applications in humans. Consequently, the reviewer recommend rejecting the current paper unless the authors can provide further clinical cases to substantiate their claims. Please strengthening the 'future perspective' section with prospects for addressing the identified issues would enhance the manuscript's quality.

  2.  
  3. 2. Clarification is needed regarding the author's definition of the terminology "CANCER." It should be explicitly stated whether the term refers exclusively to clinically malignant HUMAN cancers or encompasses various cell lines used in in vitro experiments, which might be referred to as "TUMORS." Please providing a clear and consistent definition will improve the clarity of your review article.

  4.  
  5. 3. Considering the author's publication history in PubMed, including numerous review papers in other journals, it is essential to evaluate the originality of the current article among readers in the field. Given the abundance of related review articles available on PubMed, the manuscript's original contribution must be carefully assessed as a determining factor for its publication.

Comments on the Quality of English Language

Minor correction is needed.

Round 2

Reviewer 2 Report

Comments and Suggestions for Authors

Thank you very much for your revisions. The reviewer is confident that the current manuscript is well-revised for publication in its present form.

Comments on the Quality of English Language

Minor spelling check is needed.